# Analysing the Impact of Spirulina Intake Levels on Performance Parameters, Blood Health Markers and Carcass Traits of Broiler Chickens

**DOI:** 10.3390/ani14131964

**Published:** 2024-07-02

**Authors:** Maria P. Spínola, Mónica M. Costa, José A. M. Prates

**Affiliations:** 1CIISA—Centro de Investigação Interdisciplinar em Sanidade Animal, Faculdade de Medicina Veterinária, Universidade de Lisboa, Av. da Universidade Técnica, 1300-477 Lisbon, Portugal; mariaspinola@fmv.ulisboa.pt (M.P.S.); monicacosta@fmv.ulisboa.pt (M.M.C.); 2Associate Laboratory for Animal and Veterinary Sciences (AL4AnimalS), Av. da Universidade Técnica, 1300-477 Lisbon, Portugal

**Keywords:** microalga, cumulative intake, poultry, growth performance, health status, carcass traits, meat quality

## Abstract

**Simple Summary:**

This review examines the effects of cumulative Spirulina diet intake of broilers (total feed consumed × proportion in the diet), focusing on growth performance, blood health markers and carcass quality. The analysis revealed that cumulative Spirulina intake between 14 g and 45 g per bird enhances growth performance and feed efficiency. Within this range, Spirulina supplementation also positively influences blood health markers and carcass traits, demonstrating its potential to improve overall broiler health and meat quality. However, the advantages diminish beyond this intake range, underscoring the importance of precise Spirulina dosage levels for maximum efficacy.

**Abstract:**

This systematic review examines the impact of varying Spirulina (*Limnospira platensis*) intake levels on broiler chickens, focusing on growth performance, blood health markers and carcass traits. The data revealed cumulative Spirulina intakes from 3.13 g to 521 g per bird (total feed consumed multiplied by its proportion in the diet) establish a cubic relationship between dosage and growth outcomes. Initial benefits peak and diminish with increased intake, with the optimal threshold for growth performance identified at 45 g per bird. Lower intakes between 14 g and 29 g per bird enhance blood health markers, improving lipid profiles and antioxidant capacity. Similarly, cumulative intakes of 14 g to 37 g per bird optimise meat quality, resulting in better dressing percentages, breast and thigh yields and meat tenderness while minimizing undesirable traits like abdominal fat and cooking loss. These findings underscore the importance of precisely calibrated Spirulina supplementation strategies to maximise growth, health and meat quality benefits while avoiding adverse effects at higher doses. Future research should focus on identifying optimal dosage and duration, assessing long-term implications, elucidating mechanisms of action and ensuring safety and regulatory compliance. Comparative studies with other feed additives could further establish Spirulina’s effectiveness and economic viability in poultry production.

## 1. Introduction

The search for sustainable and health-enhancing livestock feeds has become increasingly urgent in recent years, with a growing emphasis on natural additives and feedstocks that improve the overall well-being and productivity of farm animals. In this context, Spirulina (*Arthrospira platensis*, more recently proposed as *Limnospira platensis* [1]), a protein-rich microalga, has garnered attention as a potent feed supplement in poultry diets. Its potential to significantly influence the growth performance and health status of broiler chickens, in addition to its nutritional benefits, is an area of extensive research and interest [2,3].

Spirulina’s rich composition consists of essential amino acids, vitamins and antioxidants and is notably enhanced by the presence of phycocyanin, carotenoids and chlorophylls. These bioactive compounds are nutritional supplements that contribute to the growth and health improvement of broilers [4]. For instance, phycocyanin, a unique blue pigment exclusive to Spirulina, is known for its antioxidant and anti-inflammatory capabilities, essential for reducing oxidative stress and boosting immune responses in poultry [5]. Carotenoids, including beta-carotene, provide additional antioxidant benefits and are vital for immune function and visual health, whereas chlorophylls are linked to detoxification processes [6,7]. Studies reveal that Spirulina supplementation leads to marked improvements in growth rates, thanks to its high protein content and growth-promoting factors [7]. Its impact on feed efficiency further underscores Spirulina’s potential, enabling a more efficient conversion of feed into body mass, a key factor in poultry farming economics [7,8,9]. Lastly, supplementary Spirulina in broiler diets has been shown to reduce oxidative stress and improve immune responses.

The immune-enhancing properties of Spirulina stand out, with documented evidence of its role in bolstering the immune response, thereby fortifying broilers against diseases and infections [8,10]. This is attributed to Spirulina’s capacity to modulate immune functions, enhancing innate and adaptive immunity [8]. Additionally, its contribution to improved disease resistance highlights Spirulina’s utility as a natural health enhancer within poultry-production systems [5]. Moreover, the inclusion of Spirulina in broiler diets has been associated with the overall physiological well-being of chickens. The array of vitamins and minerals in Spirulina supports optimal health and vitality, improving broilers’ quality of life and welfare. The antioxidant properties of Spirulina’s bioactive compounds are crucial in this aspect, safeguarding cellular integrity and promoting healthy physiological functions [3,7]. Ongoing research into Spirulina’s effects on broiler health continues to unveil its capacity to support disease resistance and overall physiological well-being. The bioactive compounds in Spirulina not only enhance growth and feed efficiency but also fortify the immune system and contribute to the broader health status of broilers [11]. This positions Spirulina as a valuable possible addition to poultry diets, likely enhancing productivity and animal welfare [12].

However, the integration of Spirulina into broiler diets has some challenges, particularly regarding the level of its inclusion and the duration of feeding. These factors are instrumental in determining its overall impact on growth and health. While certain levels of Spirulina supplementation have shown promise in enhancing growth performance, variations in dosage and feeding duration could produce different outcomes in terms of health benefits and physiological responses [3,8,12]. This underscores the importance of identifying optimal Spirulina levels and feeding durations that effectively balance growth enhancement with health benefits. Equally important is the need to ensure the safety and regulatory compliance of the Spirulina used in poultry diets. Ensuring the absence of contaminants in Spirulina is crucial for maintaining animal health and, by extension, food safety. Compliance with regulatory standards is also essential for sustaining consumer confidence and market access [13]. The relevance of Spirulina intake on broiler meat quality concerning the various levels applied to the diet and duration of the trial was already explored in [14].

The primary aim of this review was to evaluate and synthesise scientific literature systematically available in PubMed (NCBI, Bethesda, MD, USA), Web of Science (Clarivate Analytics, Philadelphia, PA, USA), Scopus (Elsevier B.V., Amsterdam, The Netherlands) and Google Scholar (Google LLC, Mountain View, CA, USA) to ascertain the dose–response relationship between various cumulative levels of Spirulina intake and its impact on key performance parameters, blood health markers and carcass traits in broiler chickens. The cumulative microalga intake was calculated as the total feed consumed by the bird multiplied by its proportion in the diet. We hypothesised that these effects are a direct consequence of the unique transfer of Spirulina’s bioactive compounds to the birds, each following its distinct kinetics. This review endeavours to pinpoint optimal Spirulina dosage ranges that maximise growth performance, health benefits and carcass traits in broiler chickens while also identifying any potential thresholds or limits beyond which Spirulina supplementation may lead to diminishing returns or adverse effects on these parameters.

## 2. Impact of Spirulina Intake Levels on the Growth Performance of Broilers

Spirulina, known for its rich nutritional profile, has been the focus of extensive research aimed at quantifying its effects on broiler performance. Factors such as the age and initial weight of the broilers, the percentage of Spirulina in their diet, the duration of the supplementation period and cumulative intake have been studied for their impact on pivotal growth performance indicators like final body weight, cumulative weight gain, daily gain rate and feed conversion ratio [3,5,7,8,9,10,15,16,17,18,19,20,21].

Data analysis using SPSS software (version 29.0, 2024) applied various regression and curve estimation methods (linear, logarithmic, inverse, quadratic, cubic, compound, power, S, growth, exponential and logistic models), with cumulative Spirulina intake as the independent variable affecting growth metrics. Data shown in Table 1. The results (Table 2) suggest that a cubic model best describes the relationship between Spirulina dosage and broiler growth, displaying initial benefits that peak and then decline with increased intake. The cubic equations for final body weight (Y = 1617.175 + 20.066X − 0.150X^2^ + 0.000291X^3^; R^2^ = 0.170, *p* = 0.074) and cumulative body weight gain (Y = 1446.446 + 17.941X − 0.130X^2^ + 0.000239X^3^; R^2^ = 0.135, *p* = 0.171) indicate initial increases followed by a plateau, highlighting the effectiveness of low Spirulina doses. Furthermore, models for body weight daily (Y = 42.518 + 0.712X − 0.006X^2^ + 1.365 × 10^−5^X^3^; R^2^ = 0.291, *p* = 0.006) and FCR (Y = 1.790 − 0.004X + 4.239 × 10^−5^X^2^ − 9.308 × 10^−8^X^3^; R^2^ = 0.088, *p* = 0.305) underline how optimal Spirulina levels can significantly boost feed efficiency and daily growth, though higher amounts lead to diminishing returns and potential adverse effects. FBW, cumulative BWG and FCR were not statically affected by cumulative intake of Spirulina, whereas BWG was statically affected by cumulative intake. These cubic correlations between Spirulina dosage and key broiler growth indicators are illustrated in Figure 1.

The data revealed that cumulative Spirulina intake varied from 3.13 g [12] to 521 g per bird [22]. Based on the data, cumulative Spirulina intakes up to approximately 45 g per bird [17,23], corresponding to 0.5% to 1.0% of total feed over 35 to 41 days, are associated with positive growth metrics. For instance, at a cumulative intake of 17.8 g (1.0% of total feed over 27 days), broilers achieved a final body weight of 1522 g with an FCR of 1.30. Similarly, at 31.2 g (1.0% of total feed over 34 days), the final body weight was 1892 g with an FCR of 1.69. At 45.4 g (1.0% of total feed over 41 days), broilers reached a final body weight of 2872 g with an FCR of 1.60. However, the data also indicate a threshold effect, where higher cumulative Spirulina intakes do not yield additional benefits and may even reduce growth performance. For instance, at an intake of 59.5 g (8.0% of total feed over 16 days), the final body weight decreased to 1786 g, indicating a decline in performance compared to lower intake levels. Consequently, establishing the threshold for beneficial cumulative Spirulina intake at 45 g per bird appears reasonable, balancing the positive growth performance with diminishing returns at higher intake levels. It is likely that at lower cumulative Spirulina intakes, the benefits of its high protein content and improved nutrient absorption predominate, enhancing digestive efficiency. In contrast, at higher intake levels, the potential for the indigestibility of the microalga may outweigh these benefits, leading to a reduced growth performance. In summary, Spirulina intake exhibits a dose-dependent relationship with broiler growth performance, characterised by cubic dynamics. Lower cumulative intakes significantly boost growth and feed efficiency, but surpassing the optimal threshold results in diminished returns. This analysis confirms Spirulina’s nutritional value as a feed supplement and emphasises the necessity for carefully tailored dosage optimisation to leverage Spirulina’s potential fully in broiler production. The findings advocate for a balanced supplementation strategy, adapted to the specific growth phases and nutritional needs of broilers, to maximise benefits without compromising growth outcomes.

**Table 1 animals-14-01964-t001:** Impact of cumulative Spirulina intake levels on the growth performance of broilers.

Initial Age and Weight	Alga Level (% Feed) and Duration of Trial (Days) ^1^	Cumulative Alga Intake (g/Bird) ^2^	Final Body Weight (g) ^3^	Cumulative Body Weight Gain (g)	Body Weight Gain (g/d)	Feed Conversion Ratio	References
1 d old, 40.2 g	0.1%, 34 d	3.13	1994	1897	55.8	1.65	[12]
4 d old, 73.8 g	0.10%, 32 d	3.46	1863	1788	55.9	1.93	[3]
1 d old, 40.0 g	0.1%, 41 d	4.35	2527	2487	60.7	1.75 *	[15]
1 d old	0.25%, 55 d	4.58	926	891	16.2	2.08	[5]
1 d old, 40.2 g	0.2%, 34 d	6.26	2021	1923	56.6	1.63	[12]
1 d old, 41.5 g	0.25%, 34 d	7.22	-	1760	51.8	1.64	[7]
1 d old, 40.0 g	0.2%, 41 d	8.79	2681	2641	64.4	1.66 *	[15]
1 d old, 40.0 g	0.5%, 27 d	9.10	1539	1422 ^†^	52.7 ^†^	1.28	[24]
1 d old	0.5%, 55 d	9.16	932	897	16.3	2.02	[5]
22 d old, 45 g	0.5%, 20 d	13.9	2150	1231	61.6	2.03	[16]
1 d old, 41.5 g	0.5%, 34 d	14.4	-	1762	51.8	1.64	[7]
1 d old	0.5%, 35 d	15.1	2066	-	-	1.87	[17]
1 d old, 40.4 g	0.5%, 34 d	15.5	1864	-	52.1	1.71	[8]
1 d old	1.0%, 55 d	16.6	976	941	17.1	1.79	[5]
1 d old, 40.0 g	1.0%, 27 d	17.8	1522	1368 ^†^	50.7 ^†^	1.30	[24]
17 d old, 616 g	0.5%, 27 d	20.8	2664	2050	75.9	2.34	[9]
1 d old, 41.5 g	0.75%, 34 d	21.8	-	1787	52.6	1.63	[7]
21 d old	4.0%, 16 d	23.1	1867	-	-	-	[25]
1 d old, 40.0 g	1.5%, 27 d	28.1	1604	1452 ^†^	53.8 ^†^	1.29	[24]
1 d old, 42.0 g	1.0%, 34 d	28.1	-	1771	52.1	1.59	[18]
22 d old, 45 g	1.0%, 20 d	28.9	2210	1384	69.2	1.90	[16]
1 d old, 41.5 g	1.0%, 34 d	28.9	-	1801	53.0	1.61	[7]
1 d old	1.0%, 35 d	30.2	2162	-	-	1.72	[17]
1 d old, 40.4 g	1.0%, 34 d	31.2	1892	-	52.9	1.69	[8]
1 d old, 43 g	1.0%, 41 d	37.2	-	2248	54.8	1.65	[10]
17 d old, 616 g	1.0%, 27 d	40.1	2729	2054	76.1	2.06	[9]
22 d old, 45 g	1.5%, 20 d	40.4	2140	1241	62.1	1.94	[16]
1 d old	1.5%, 35 d	45.2	1922	-	-	1.90	[17]
1 d old, 40.0 g	1.0%, 41 d	45.4	2872	2832	69.1	1.60	[23]
21 d old	8.0%, 16 d	46.3	1786	-	-	-	[25]
1 d old	2.0%, 34 d	65.4	2138	1993	58.6	1.64	[26]
0 d old	3.0%, 35 d	79.7	1805	1740	49.7	1.50	[27]
21 d old	15%, 14 d	84.7	1697	1039	74.2	1.63	[28]
17 d old, 616 g	2.0%, 27 d	85.9	279	2136	79.1	2.30	[9]
1 d old, 40.0 g	2.0%, 41 d	90.5	2889	2850	69.5	1.58	[23]
8 d old	3.0%, 34 d	116	2311	2136	62.8	1.80	[19]
0 d old, 36 g	2.5%, 49 d	130	3070	2940	60.0	1.66	[20]
1 d old, 40.0 g	3.0%, 41 d	135	2861	2819	68.8	1.59	[23]
1 d old, 40.0 g	4.0%, 41 d	180	2933	2893	70.6	1.55	[23]
1 d old, 47.3 g	10.8%, 34 d	211	1063	1040 ^†^	30.6 ^†^	1.89	[29]
7 d old, 135 g	15%, 28 d	221	998	902	32.2	1.78	[30]
14 d old, 351 g	15%, 21 d	230	1183	832	39.6	1.91	[31]
15 d old ^4^	10%, 20 d	254	2091	1361	68.0	1.81	[32]
8 d old	6.0%, 34 d	256	2454	2281	67.1	1.82	[19]
15 d old ^5^	10%, 22 d	315	2372	1851	84.1	1.68	[32]
1 d old	10.8%, 34 d	324	2121	-	-	-	[13]
1 d old	17.3%, 34 d	521	2260	-	-	-	[22]

^1^ For the duration of the trial, the last day corresponding to slaughtering was not considered. ^2^ Calculated as the total feed ingested per animal during the experimental period multiplied by the proportion of microalgae in the diet. For some of the studies, no information about the cumulative feed intake (CFI) was available and, therefore, an estimation of that was conducted as follows: CFI (g/bird) (Altmann et al. [13]; Altmann et al. [22]; Shanmugapriya et al. [17]) = (CFI (Abdel-Moneim et al. [8]) + CFI (Ibrahim et al. [26]) + CFI (Neumann et al. [29]) + CFI (Park et al. [7]) + CFI (Sugiharto et al. [18]))/5; the studies had the same duration of trial and animal initial age; the trial from Shanmugapriya et al. [17] lasted 1 more day but we performed the same calculation; CFI (g/bird) (Moustafa et al. [16]) = CFI (g/d/bird) × number of days; CFI (g/bird) (Mullenix et al. [32]) = CFI (lb/bird) × 453.59237; CFI (g/bird) (Costa et al. [31]; Pestana et al. [28]; Spínola et al. [30]) = (CFI (g/d/pen) × number of days)/number of birds; CFI (g/bird) (Toyomizu et al. [25]) = CFI (Pestana et al. [28]) + 2× (CFI (Pestana et al. [28])/14). In the study by Feshanghchi et al. [10], the basal diet contained aflatoxin B1 (0.6 mg/kg) whereas, in the report by Mullenix et al. [32], the basal diet had a low crude protein level. ^3^ Final body weight: adding the initial age to the duration of the trial (days) equals the age of the birds. ^4^ Female broilers. ^5^ Male broilers. * Estimated feed conversion ratio (FCR) = Cumulative feed intake/Body weight gain. ^†^ Estimated cumulative body weight gain (BWG) (g) = Cumulative feed intake/feed conversion ratio (FCR); BWG (g/d) = Cumulative BWG (g)/duration of trial (days).

**Table 2 animals-14-01964-t002:** Summary of correlation analysis for predicting performance-related dependent variables based on cumulative Spirulina intake.

Variable	Best Model Type	R-Square	Degrees of Freedom	*p*-Value	Model Equation
Final body weight	Cubic	0.170	37	0.073	Y = 1617.175 + 20.066X − 0.150X^2^ + 0.000291X^3^
Cumulative body weight gain	Cubic	0.135	34	0.171	Y = 1446.446 + 17.941X − 0.130X^2^ + 0.000239X^3^
Body weight gain	Cubic	0.291	36	0.006	Y = 42.518 + 0.712X − 0.006X^2^ + 1.365 × 10^−5^X^3^
Feed conversion ratio	Cubic	0.088	39	0.305	Y = 1.790 − 0.004X + 4.239 × 10^−5^X^2^ − 9.308 × 10^−8^X^3^

## 3. Influence of Spirulina Intake Levels on the Blood Health Indicators of Broilers

Table 3 provides an in-depth analysis of the effects of various cumulative Spirulina intake levels on essential blood health markers in broilers, such as triglycerides, total cholesterol, HDL cholesterol, LDL cholesterol, superoxide dismutase (SOD), glutathione (GSH), malondialdehyde (MDA) and total antioxidant capacity (TAC). This detailed exploration highlights Spirulina’s potential to modulate blood lipid profiles and bolster antioxidant defences, offering valuable insights into its broader health benefits.

The correlation analyses detailed in Table 4 offer a more nuanced view, revealing that the relationship between Spirulina dosage and health marker enhancement is intricate and nonlinear. Models such as (Y = 102.010 + (243.629/X)) for triglycerides and (Y = 141.513 + (43.592/X)) for cholesterol show inverse relationships, suggesting diminishing returns with higher Spirulina intakes. Similarly, antioxidant markers like SOD and GSH, modelled through cubic and quadratic functions, respectively, display dose-dependent responses that suggest a beneficial range beyond which Spirulina’s efficacy may plateau or decline. However, the correlation analysis for GSH and TAC have a low number of degrees of freedom and, therefore, the output should be interpreted with care. Furthermore, the variability in how Spirulina influences LDL and HDL cholesterol levels, crucial for cardiovascular health, emphasises the complexity of Spirulina’s effects on lipid metabolism.

The data span a range of cumulative Spirulina intakes from 3.13 g to 521 g per bird, with significant variability in health markers. Notably, lower Spirulina intake frequently resulted in positive lipid profile changes, including decreased triglyceride and cholesterol levels. For example, research by Hassan et al. [5] observed reductions in these markers at intakes of 4.58 g and 9.16 g, affirming Spirulina’s effectiveness in enhancing lipid metabolism. Further evidence from Elbaz et al. [12] and Moustafa et al. [16] indicates increased levels of antioxidant markers such as SOD, which suggests that Spirulina also plays a crucial role in strengthening antioxidant defences and potentially mitigating oxidative stress.

Based on the analysis of health markers in broilers’ blood, it appears that moderate cumulative Spirulina intakes of around 13.9 g to 28.9 g per bird are optimal for improving various health parameters. These intake levels, corresponding to approximately 0.5% to 1.0% of total feed over the respective trial periods, were associated with beneficial outcomes such as higher HDL levels, increased SOD and enhanced TAC. For instance, at an intake of 13.9 g, HDL levels were 65.9 mg/dL, SOD was 6.30 U/mL and TAC was 7.20 U/mL. Similarly, at 28.9 g, HDL levels reached 70.3 mg/dL and TAC increased to 8.10 U/mL. These moderate intakes also helped maintain lower levels of triglycerides and LDL. Therefore, cumulative Spirulina intakes of 14 g to 29 g per bird are recommended for optimal health benefits in broilers.

At lower cumulative Spirulina intakes (14–29 g per bird), the benefits on blood health markers are likely primarily due to its bioactive compounds like phycocyanin, carotenoids and essential fatty acids, which improve lipid profiles and enhance antioxidant capacity. These compounds lower triglycerides and LDL cholesterol while increasing HDL cholesterol and boosting antioxidant enzymes like superoxide dismutase (SOD) and glutathione (GSH), reducing oxidative stress. At higher intakes (exceeding 45 g per bird), the benefits diminish due to potential indigestibility and nutrient overload, which can reduce the efficiency of nutrient absorption and increase the risk of accumulating harmful contaminants, thereby negating some of the positive effects seen at lower levels.

This analysis also acknowledges the impact of environmental factors, such as heat stress, which can skew baseline health markers and confound Spirulina’s effects, as seen in the elevated triglyceride levels under such conditions in studies by Elbaz et al. [12] and Moustafa et al. [16].

In conclusion, while Spirulina demonstrates a potentially beneficial dietary supplement for broilers, optimising the dosage is critical. The findings from Table 4 stress the importance of identifying effective Spirulina levels to maximise health benefits without overreaching a threshold where additional intake is ineffectual. This complex dose–response relationship underscores the need for ongoing research to refine supplementation strategies, ensuring Spirulina’s nutritional benefits are fully leveraged for broiler health and production.

## 4. Influence of Spirulina Intake Levels on Broilers’ Carcass Traits

Table 5 provides a comprehensive analysis of how varied cumulative Spirulina (*Limnospira platensis*) intake levels influence broiler carcass traits, encompassing dressing percentage, breast and thigh yield, abdominal fat percentage, shear force, cooking loss, water holding capacity and drip loss. This in-depth examination sheds light on the subtle yet significant impact of Spirulina incorporation in the diet on both carcass composition and overall meat quality.

The correlation analyses detailed in Table 6 utilise various modelling approaches to predict carcass-related outcomes based on Spirulina intake levels. For instance, the dressing percentage is modelled using a power function (Y = 67.009X^0.022^), indicating a modest correlation (R^2^ = 0.121, *p* = 0.088). In contrast, breast yield follows a cubic relationship (Y = 19.281 + 0.501X − 0.003 X^2^ + 5.714 × 10^−6^ × X^3^), suggesting a strong correlation (R^2^ = 0.520, *p* = 0.020) and highlighting how initial increases in Spirulina intake enhance carcass yields before reaching a point of diminishing returns. The correlation analysis of shear force, water holding capacity and drip loss have a low number of degrees of freedom and, as explained before, the output should be interpreted with care.

The dataset spans a broad range of Spirulina intakes, from 3.46 g to 521 g per bird, showing notable variations in carcass traits. Specifically, lower intake levels consistently yield marked improvements in carcass quality. For instance, Elbaz et al. [12] observed a dressing percentage of 72.1% at just 3.13 g of intake, indicating an enhanced carcass yield at lower supplementation levels. Further research by Park et al. [7] and Moustafa et al. [16] elaborates on how Spirulina influences meat composition, notably affecting breast and thigh yields and abdominal fat percentages. For example, Park et al. [7] observed that breast yields were 19.1% and 19.6% at cumulative intakes of 7.22 g and 21.8 g, respectively, underscoring Spirulina’s role in improving meat quality attributes. However, as Spirulina intake increases, its relationship with carcass traits becomes increasingly complex, illustrated by a plateau in benefits such as those reported by Sugiharto et al. [18], who noted a dressing percentage of 72.8% and a breast yield of 35.6% at 28.1 g of intake. This suggests that while Spirulina can enhance certain carcass characteristics, there is a threshold beyond which no further improvements are observed.

Based on the analysis of carcass traits in broilers, moderate cumulative Spirulina intakes of around 13.9 g to 37.2 g per bird are optimal for enhancing meat quality. These intake levels, corresponding to approximately 0.5% to 1.0% of total feed over the respective trial periods, yield positive outcomes in various meat quality parameters. For example, at 13.9 g, broilers exhibited a dressing percentage of 68.5%, a breast percentage of 37.9% and a shear force of 1.61 kg, indicating tender meat. Similarly, at 37.2 g, the breast percentage was 40.3% and the thigh percentage was 23.7%, with a shear force of 1.05 kg, further highlighting improved meat tenderness. Additionally, these moderate intakes help maintain low abdominal fat percentages and reduce cooking and drip losses, resulting in better overall meat quality. Therefore, cumulative Spirulina intakes from 14 g to 37 g per bird are recommended for optimal meat quality in broilers.

At lower cumulative Spirulina intakes (14–37 g per bird), the positive effects on carcass traits, such as dressing percentage, breast and thigh yields and meat quality, are likely primarily due to Spirulina’s high protein content and rich array of bioactive compounds. The high-quality protein in Spirulina supports muscle growth and development, leading to improved carcass yields. Bioactive compounds like phycocyanin and carotenoids enhance antioxidant defences, reducing oxidative stress and improving meat tenderness and water-holding capacity. Additionally, Spirulina’s prebiotic properties promote beneficial gut microflora, enhancing nutrient absorption and digestive efficiency. However, at higher intake levels (exceeding 45 g per bird), these benefits diminish due to potential issues with digestibility and nutrient overload, which can lead to inefficient nutrient absorption and increased metabolic strain, thereby reducing the positive impacts on carcass traits.

In summary, the analysis demonstrates that low levels of Spirulina supplementation are generally associated with improved carcass traits and potentially enhanced meat quality. Nevertheless, the complex interactions between Spirulina intake and carcass composition, particularly at higher intake levels, necessitate further research. Future investigations should not only aim to optimise supplementation strategies but also consider the broader implications for meat quality and consumer acceptance. This approach will ensure that the benefits derived from Spirulina are maximised without plateauing or declining at excessive doses, thereby maintaining the balance between nutritional enhancement and consumer satisfaction.

## 5. Safety Precautions and Regulatory Aspects of Spirulina Usage in Broilers Diets

In assessing the safety precautions and regulatory aspects related to the use of Spirulina as a feed additive or ingredient in broiler diets, several critical considerations come to the forefront. First, the safety of dietary Spirulina is generally acknowledged, particularly when it is free from contaminants. Spirulina is generally considered safe by the European Food Safety Authority and previous studies [33]. Indeed, they have found that various contaminants from freshwater are typically present in Spirulina below detectable levels, reinforcing its safety profile. Grosshagauer et al. [34] also observed that mercury, aluminium, cadmium and arsenic in previously analysed Spirulina samples were within acceptable intake levels, although some samples showed slightly elevated levels of lead, albeit in less than half of the samples tested. Studies such as El-Bahr et al. [3] emphasise the importance of ensuring Spirulina is free from contaminants like heavy metals or harmful microorganisms, which can pose significant health risks to both the poultry and, subsequently, to consumers. The potential for bioaccumulation of these contaminants in broiler tissues, especially with higher levels of Spirulina intake, necessitates rigorous quality controls and regular safety assessments. This aspect underscores the need for well-established safety protocols in the production and processing of Spirulina intended for animal feed since Spirulina is generally considered safe when it is pure and well-processed, meaning stringent quality control measures are crucial. Although microcystins can be detected in Spirulina used as fish feed supplements, proper cultivation and production conditions can ensure levels below 1 µg/L, which are monitored through toxicity analysis of the microalgae [18,33]. Additionally, Choi et al. [35] identified the presence of certain bacteria, including *Leucobacter* sp., *Aeromicrobium* sp., *Staphylococcus* spp. and *Halomonas* spp., isolated from *A. platensis*. The first three genera were suggested to originate from human skin and may contaminate Spirulina during the subculturing process, while contamination with *Halomonas* spp. could stem from collected water samples.

Additionally, the regulatory landscape surrounding the use of Spirulina in animal feed is complex and varies across different regions. As Altmann et al. [13] discuss, compliance with local and international regulations concerning feed safety, permissible additive levels and labelling requirements is paramount. These regulations are designed to ensure the safety of animal feed additives and, by extension, the safety of animal-derived food products for human consumption. Furthermore, the regulatory standards often evolve in response to new scientific findings and public health considerations. For instance, the studies conducted by Pestana et al. [28] and others contribute to a growing body of evidence that regulators may use to review and update guidelines on the use of Spirulina in poultry diets.

The long-term safety of Spirulina, particularly at high inclusion levels and over extended feeding durations, remains an area requiring further research. While short-term studies indicate beneficial effects, the long-term implications for animal and human health are not fully understood. This knowledge gap calls for ongoing research and monitoring to detect any potential adverse effects, including the cumulative impact of bioactive compounds in Spirulina on animal health and food safety.

In summary, while Spirulina offers potential health benefits as a poultry feed additive, its safe inclusion in broiler diets demands a comprehensive approach encompassing rigorous quality control, adherence to evolving regulatory standards and continuous research into its long-term safety and efficacy. Such an approach is essential to ensure that Spirulina-enhanced broiler meat is not only beneficial but also safe and compliant with regulatory requirements, thereby maintaining consumer trust and market viability.

Figure 2 summarises the main outputs of dietary Spirulina incorporation in broiler diets reviewed in this work.

## 6. Conclusions and Future Research

This review synthesises findings on the effects of Spirulina incorporation in broiler chickens’ diets, revealing a range of cumulative microalga intake from 3.13 g to 521 g per bird. A cubic relationship between Spirulina dosage and key growth outcomes was established, with the threshold for beneficial cumulative Spirulina intake defined at 45 g per bird. This threshold balances positive growth performance with diminishing returns observed at higher intake levels, as higher doses in the diet do not produce additional gains and may have detrimental effects. The relationship between Spirulina dosage and blood health markers is intricate and nonlinear. Optimal health benefits in broilers are observed with cumulative Spirulina intakes of 14 g to 29 g per bird. This range is associated with improved lipid profiles, increased antioxidant capacity and reduced oxidative stress.

Similarly, the relationship between Spirulina dosage and meat quality enhancement is not linear. Optimal meat quality in broilers is achieved with cumulative Spirulina intakes of 14 g to 37 g per bird. This range results in improved dressing percentages, breast and thigh meat yields and meat tenderness while minimizing undesirable traits such as abdominal fat and cooking loss. These findings underscore the importance of carefully calibrating Spirulina dosages to maximise its benefits while ensuring safety, highlighting the challenge of determining optimal levels that balance effectiveness with potential risks due to its complex dose–response relationship.

Future research should focus on several key areas to maximise Spirulina’s utility in poultry nutrition effectively. Identifying the optimal dosage and duration of Spirulina supplementation is paramount to ensure enhanced growth and health benefits without adverse effects. Longitudinal studies are critical to assess the long-term implications of Spirulina use on broiler health and meat quality, including the potential accumulation of bioactive compounds. Additionally, elucidating the mechanisms by which Spirulina influences broiler physiology will aid in refining supplementation strategies for targeted outcomes.

Addressing Spirulina’s safety and compliance with regulatory standards is also essential. Rigorous quality control is necessary to prevent contaminant risks and adherence to regulatory guidelines must be ensured to maintain consumer confidence in Spirulina-supplemented poultry products. Moreover, comparative studies with other feed additives could offer insights into Spirulina’s relative effectiveness and economic viability.

## Figures and Tables

**Figure 1 animals-14-01964-f001:**
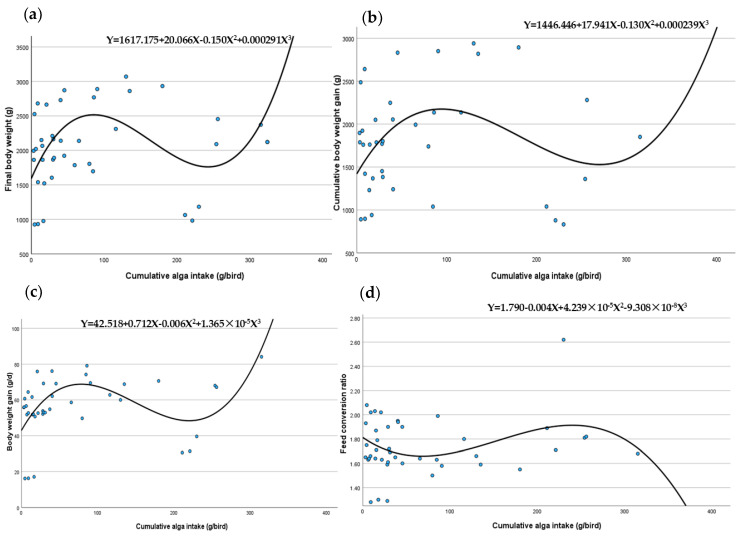
The relationship among Spirulina dosage and key broiler growth indicators is characterised by cubic models: (**a**) final body weight; (**b**) cumulative body weight gain; (**c**) body weight gain; (**d**) feed conversion ratio.

**Figure 2 animals-14-01964-f002:**
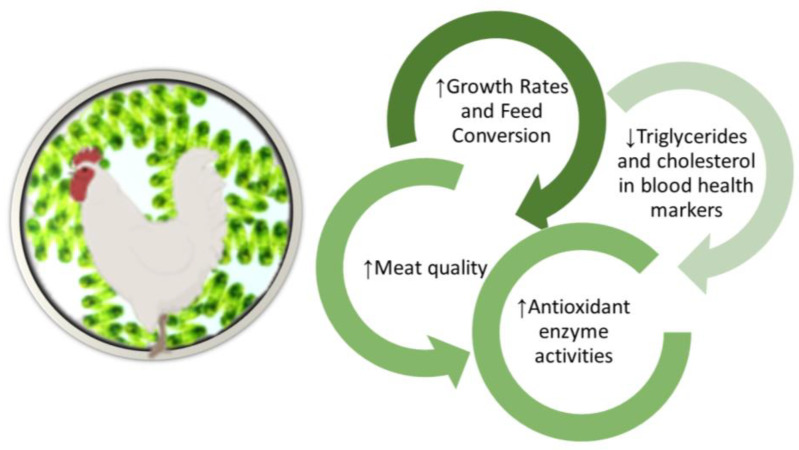
Summary of the effect of Spirulina intake on performance parameters, blood health markers and carcass traits of broiler chickens.

**Table 3 animals-14-01964-t003:** Impact of cumulative Spirulina intake levels on health markers in the blood of broilers.

Initial Age and Weight	Alga Level (% Feed) and Duration of Trial (Days) ^1^	Cumulative Alga Intake (g/Bird) ^2^	Lipid Profile				Antioxidants				References ^†^
			Triglyceride (mg/dL)	Total Cholesterol (mg/dL)	HDL Cholesterol (mg/dL)	LDL Cholesterol (mg/dL)	SOD ^3^ (U/mL)	GSH ^4^ (mol/L)	MDA ^5^ (nmol/mL)	TAC ^4^ (U/mL)	
1 d old, 40.2 g	0.1%, 34 d	3.13	223	188	46.5	97.9	145	-	0.76	-	[12]
1 d old	0.25%, 55 d	4.58	60.2	1.01	-	-	-	-	-	-	[5]
1 d old, 40.2 g	0.2%, 34 d	6.26	226	194	50.8	99.2	145		0.81	-	[12]
1 d old, 41.5 g	0.25%, 34 d	7.22	-	-	-	-	5.50	-	-	-	[7]
1 d old, 40.0 g	0.5%, 27 d	9.10	-	119	-	-	-	-	-	-	[24]
1 d old	0.5%, 55 d	9.16	68.0	93.5	-	-	-	-	-	-	[5]
22 d old, 45 g	0.5%, 20 d	13.9	176	201	65.9	100	6.30	29.7	3.10	7.20	[16]
1 d old, 41.5 g	0.5%, 34 d	14.4	-	-	-	-	5.80	-	-	-	[7]
1 d old	1.0%, 55 d	16.6	63.9	90.9	-	-	-	-	-	-	[5]
1 d old, 52.5 g	0.5%, 34 d	16.7	79.0	156	58.5	61.3	-	-	-	-	[21]
1 d old, 40.0 g	1.0%, 27 d	17.8		117		-	-	-	-	-	[24]
17 d old, 616 g	0.5%, 27 d	20.8	84.0	141	60.3	-	80.2		11.1		[9]
1 d old, 41.5 g	0.75%, 34 d	21.8	-	-	-	-	5.90	-	-	-	[7]
1 d old, 40.0 g	1.5%, 27 d	28.1	-	132	-	-	-	-	-	-	[24]
1 d old, 42.0 g	1.0%, 34 d	28.1	127	139	64.5	61.6	-	-	-	-	[18]
22 d old, 45 g	1.0%, 20 d	28.9	175	191	70.3	85.2	6.10	34.6	2.70	8.10	[16]
1 d old, 41.5 g	1.0%, 34 d	28.9	-	-	-	-	6.20	-	-	-	[7]
1 d old, 52.5 g	1.0%, 34 d	35.1	74.0	146	57.5	65.1	-	-	-	-	[21]
1 d old, 43 g	1.0%, 41 d	37.2	104	142	-	-	-	-	-	-	[10]
17 d old, 616 g	1.0%, 27 d	40.1	77.1	138	64.1	-	79.9	-	10.0	-	[9]
22 d old, 45 g	1.5%, 20 d	40.4	175	187	73.1	78.5	5.80	32.1	2.80	8.60	[16]
21 d old	15%, 14 d	84.7	142	148	105	14.1	-	-	-	-	[28]
17 d old, 616 g	2.0%, 27 d	85.9	76.9	127	53.2	-	90.3		9.54	-	[9]
8 d old	3.0%, 34 d	116	119	145	62.5	77.5	-	-	-	-	[19]
8 d old	6.0%, 34 d	256	92.6	139	64.4	69.5	-	-	-	-	[19]

^1^ For the duration of the trial, the last day corresponding to slaughtering was not considered. ^2^ Calculated as the total feed ingested per animal during the experimental period multiplied by the proportion of microalgae in the diet. For some of the studies, no information about the cumulative feed intake (CFI) was available and, therefore, an estimation was conducted as follows: CFI (g/bird) (Moustafa et al. [16]) = CFI (g/d/bird) × number of days; CFI (g/bird) (Pestana et al. [28]) = (CFI (g/d/pen) × number of days)/number of birds. In the study by Elbaz et al. [12], the broilers were kept at standard rearing conditions until day 10, and then the ambient temperature was increased from 29 °C to 34 ± 2 °C for 12 h for three consecutive days a week until 35 days of age. In the report by Moustafa et al. [16], the animals were exposed to cyclic heat stress at 34 ± 1 °C for 8 h per day, whereas in the study by Mirzaie et al. [9], broilers were exposed to heat stress at 36 °C for 6 h/day from day 38 to 44. ^3^ Obtained using a commercial kit from Cayman Chemical Company (Cayman Chemical Co., Ann Arbor, MI, USA), where one unit was defined as the amount of enzyme needed to exhibit 50% dismutation of the superoxide radical measured in a change in absorbance per minute at 25 °C and pH = 8.0 (Park et al. [7]); one unit of the enzyme was defined as the amount of enzyme capable of inhibiting the rate of pyrogallol oxidation by 50% (Mirzaie et al. [9]). ^4^ Determined following manufacturer’s indications from a commercial kit from Nanjing Jianheng Bioengineering Institute (Nanjing, Jiangsu, China). ^5^ Measured with an ELISA kit from QuantiChrom (BioAssay Systems, USA, and Cayman Chemical Company, USA) (Elbaz et al. [12]) or by the thiobarbituric acid reaction method (Mirzaie et al. [9]; Moustafa et al. [16]) with a commercial colourimetric assay kit (Moustafa et al. [16]). Abbreviations: HDL, high-density lipoprotein; LDL, low-density lipoprotein; SOD, superoxide dismutase; GSH, glutathione; MDA, malondialdehyde; TAC, total antioxidant capacity. ^3,5^ Oxidative stress markers. ^†^ Reference [7] is under normal conditions, while references [9,12,16] are under stress conditions (heat stress and high ambient temperature).

**Table 4 animals-14-01964-t004:** Summary of correlation analysis for predicting health-related dependent variables based on cumulative Spirulina intake.

Variable	Best Model Type	R-Square	Degrees of Freedom	*p*-Value	Model Equation
Triglycerides	Inverse	0.140	16	0.126	Y = 102.010 + (243.629/X)
Total cholesterol	Inverse	0.011	19	0.650	Y = 141.513 + (43.592/X)
HDL cholesterol	S-curve	0.338	12	0.029	Y = 100/(1 + e^−0.5(Dosage−50)^)
LDL cholesterol	Cubic	0.513	7	0.148	Y = 110.486 − 1.943X + 0.018X^2^ − 4.286 × 10^−5^X^3^
SOD	Cubic	0.481	8	0.136	Y = 168.215 − 14.969X + 0.414X^2^ − 0.003X^3^
GSH	Quadratic (also cubic)	1.000	0 *	-	Y = 16.912 + 1.205X − 0.021X^2^
MDA	Power	0.656	6	0.015	Y = 0.304 × X0.803
TAC	Linear	0.992	1 *	0.056	Y = 6.492 + 0.053X

* Low number of degrees of freedom. Abbreviations: HDL, high-density lipoprotein; LDL, low-density lipoprotein; SOD, superoxide dismutase; GSH, glutathione; MDA, malondialdehyde; TAC, total antioxidant capacity.

**Table 5 animals-14-01964-t005:** Impact of cumulative Spirulina intake levels on carcass traits of broilers.

Initial Age and Weight	Alga Level (% Feed) and Duration of Trial (Days) ^1^	Cumulative Alga Intake (g/Bird) ^2^	Dressing (%)	Breast (%) ^3^	Thigh (%) ^3^	Abdominal Fat (%)	Shear Force (kg) ^4^	Cooking Loss (%)	Water Holding Capacity (%)	Drip Loss (%)	References
1 d old, 40.2 g	0.1%, 34 d	3.13	72.1	-	-	0.59	-	-	-	-	[12]
4 d old, 74.1 g	0.10%, 32 d	3.46	-	-	-	-		12.4	78.9		[3]
1 d old, 40.0 g	0.1%, 41 d	4.35	62.4	-	-	2.53	-	-	-	-	[15]
1 d old	0.25%, 55 d	4.58	69.1	-	-	-	-	-	-	-	[5]
1 d old, 40.2 g	0.2%, 34 d	6.26	71.9	-	-	0.57	-	-	-	-	[12]
1 d old, 41.5 g	0.25%, 34 d	7.22	-	19.1	-	2.91	-	30.0	39.5	13.1	[7]
1 d old, 40.0 g	0.2%, 41 d	8.79	64.3	-	-	2.22	-	-	-	-	[15]
1 d old, 40.0 g	0.5%, 27 d	9.10	68.7		-		-	-	-	-	[24]
1 d old	0.5%, 55 d	9.16	69.5		-	-	-	-	-	-	[5]
22 d old, 45 g	0.5%, 20 d	13.9	68.5	37.9	-	-	1.61	13.5	-	-	[16]
1 d old, 41.5 g	0.5%, 34 d	14.4	-	19.1	-	2.85	-	29.6	39.2	12.9	[7]
1 d old	1.0%, 55 d	16.6	69.3	-	-		-	-	-	-	[5]
1 d old, 52.5 g	0.5%, 34 d	16.7	92.2	-	-		-	-	-	-	[21]
1 d old, 40.0 g	1.0%, 27 d	17.8	71.5	-	-		-	-	-	-	[24]
1 d old, 41.5 g	0.75%, 34 d	21.8	-	19.6	-	2.81	-	28.7	38.4	12.9	[7]
21 d old	4.0%, 16 d	23.1	-	-	-	1.26	-	-	-	-	[25]
1 d old, 40.0 g	1.5%, 27 d	28.1	71.7	-	-	-	-	-	-	-	[24]
1 d old, 42.0 g	1.0%, 34 d	28.1	72.8	35.6	15.5	1.99	-	-	-	-	[18]
22 d old, 45 g	1.0%, 20 d	28.9	69.0	38.5	-	-	1.55	12.3			[16]
1 d old, 41.5 g	1.0%, 34 d	28.9	-	19.7	-	2.70	-	28.5	37.6	12.8	[7]
1 d old, 52.5 g	1.0%, 34 d	35.1	88.4	-	-	-	-	-	-	-	[21]
1 d old, 43 g	1.0%, 41 d	37.2	60.2	40.3	23.7	1.05	-	-	-	-	[10]
22 d old, 45 g	1.5%, 20 d	40.4	68.0	37.0	-	-	1.58	12.6	-	-	[16]
1 d old, 40.0 g	1.0%, 41 d	45.4	76.0	40.0	13.4	-	-	-	-	-	[23]
21 d old	8.0%, 16 d	46.3	-	-	-	1.29	-	-	-	-	[25]
0 d old	3.0%, 35 d	79.7	-	29.1	-	-	-	-	-	-	[27]
21 d old	15%, 14 d	84.7	-	-	-	-	1.53	12.6	-	-	[28]
1 d old, 40.0 g	2.0%, 41 d	90.5	76.1	40.8	13.1	-	-	-	-	-	[23]
1 d old, 40.0 g	3.0%, 41 d	135	77.6	39.6	12.9	-	-	-	-	-	[23]
1 d old, 40.0 g	4.0%, 41 d	180	75.6	38.7	13.2	-	-	-	-	-	[23]
15 d old ^5^	10%, 20 d	254	74.0	18.4	12.4	1.34	-	-	-	-	[32]
15 d old ^6^	10%, 22 d	315	75.0	20.3	13.2	1.38	-	-	-	-	[32]
1 d old	10.8%, 34 d	324	74.4	20.0	-	-	1.14	29.0			[13]
1 d old	17.3%, 34 d	521	75.2	-	-	-	1.04	25.9		1.86	[22]

^1^ For the duration of the trial, the last day corresponding to slaughtering was not considered. ^2^ Calculated as the total feed ingested per animal during the experimental period multiplied by the proportion of microalgae in the diet. For some of the studies, no information about the cumulative feed intake (CFI) was available and, therefore, an estimation that was conducted as follows: CFI (g/bird) (Altmann et al. [13]; Altmann et al. [22]) = (CFI (Abdel-Moneim et al. [8]) + CFI (Ibrahim et al. [26]) + CFI (Neumann et al. [29]) + CFI (Park et al. [7]) + CFI (Sugiharto et al. [18]))/5; the studies had the same duration of trial and animal initial age. CFI (g/bird) (Moustafa et al. [16]) = CFI (g/d/bird) × number of days; CFI (g/bird) (Mullenix et al. [32]) = CFI (lb/bird) × 453.59237; CFI (g/bird) (Pestana et al. [28]) = (CFI (g/d/pen) × number of days)/number of birds; CFI (g/bird) (Toyomizu et al. [25]) = CFI (Pestana et al. [28]) + 2× (CFI (Pestana et al. [28])/14). ^3^ The breast and thigh values (%) presented by Mullenix et al. [32] were converted from % of final body weight to % of carcass weight as follows: % breast/thigh relative to carcass weight = (% breast/thigh relative to body weight × carcass weight)/body weight. The total breast values reported by Mullenix et al. [32] corresponded to breast yield plus tender yield (%). The results presented by Sugiharto et al. [18] are expressed as % of eviscerated carcass, and the ones described by Feshanghchi et al. [10] and Abbass et al. [25] as % of carcass weight. ^4^ Shear force (kg) (Altmann et al. [13]; Altmann et al. [22]) = Shear force (N) × 0.1019716. ^5^ Female broilers. ^6^ Male broilers.

**Table 6 animals-14-01964-t006:** Summary of correlation analysis for predicting carcass-related dependent variables based on cumulative Spirulina intake.

Variable	Best Model Type	R-Square	Degrees of Freedom	*p*-Value	Model Equation
Dressing	Power	0.121	23	0.088	Y = 67.009 × X^0.022^
Breast	Cubic	0.520	13	0.020	Y = 19.281 + 0.501X − 0.003X^2^ + 5.714 × 10^−6^X^3^
Thigh	Power	0.633	6	0.018	Y = 42.234X^−0.252^
Abdominal fat	S-curve	0.080	12	0.327	Y = 0.620/(1 + e^−(−1.671X)^)
Shear force	Cubic	0.995	2 *	0.007	Y = 1.599 + 0.000X − 5.997 × 10^−6^X^2^ + 9.147 × 10 − 9X^3^
Cooking loss	Cubic	0.233	7	0.576	Y = 24.461 − 0.214X + 0.001X^2^ − 1.445 × 10^−6^X^3^
Water holding capacity	S-curve	0.886	3 *	0.017	Y = 3.467/(1 + e^−2.862X^)
Drip loss	Linear (also cubic and quadratic)	1.000	3 *	0.000	Y = 13.321 − 0.022X

* Low number of degrees of freedom.

## Data Availability

The data presented in this study are available upon request from the corresponding author.

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
