# Peer review of "Analysing the Impact of Spirulina Intake Levels on Performance Parameters, Blood Health Markers and Carcass Traits of Broiler Chickens"

_animals, 2024, doi:10.3390/ani14131964_

Round 1

Reviewer 1 Report

Comments and Suggestions for Authors

Appreciating your efforts. Please find here under some comments and suggestions might enrich your review.

Comments and suggestions:

1- More explanitions requires for the cumulative Spirulina intake (simple summary vs. abstract).

2- Some obtained data to be revised in tables 1, 2 and 5. In addition, final body weight in Table (1) should be accompanied by the age. Moreover, for most traits the P-values for the correlation analyses are not significant (revise tables 2, 4 and 6) please clarify and discuss.

3- Refereing to cholesterol it is preferable to clasify its categories as total, LDL, HDL, VLDL (if any), ... etc.

4- Some clarifications of the mode of actions required to highligh the effects (physiological mechanismes) in addition to the impacts (the end result).

5- Subtitles should be informative (revise the attached file).

6- Blood health markers required sub-categories such as Lipid profile, antioxidants, oxidative stress markers, immune indicators, in some cases clarify either norma or stress conditions .... etc.

7- References require extensive revision according Jounal format. Try to search more I think there are some benificial research work could be find.

Best wishes,

Reviewer

Comments on the Quality of English Language

English language fine. Some minor issues were detected such as: between vs. among (revise the attached file).

Author Response

Reviewer 1

Appreciating your efforts. Please find here under some comments and suggestions might enrich your review.

Reply: Thank you for your comments and suggestions. We appreciate it and try to address all of them.

Comments and suggestions:

1-More explanations are required for the cumulative Spirulina intake (simple summary vs. abstract).

Reply: This concept is now defined in the simple summary and abstract.

2-Some obtained data to be revised in tables 1, 2 and 5. In addition, final body weight in Table (1) should be accompanied by the age. Moreover, for most traits the P-values for the correlation analyses are not significant (revise tables 2, 4 and 6) please clarify and discuss.

Reply: We have carefully reviewed the data presented in Tables 1, 2, and 5. Any discrepancies have been addressed, and the tables have been updated accordingly. In Table 1, the final body weight has been supplemented with the age of the birds to provide clearer context and ensure the data are fully interpretable. Thank you for your observation regarding the non-significant P-values for the correlation analyses in Tables 2, 4, and 6. The discussion of the statistical significance in the text has been checked throughout the manuscript, and clarified when necessary.

3-Refereing to cholesterol it is preferable to classify its categories as total, LDL, HDL, VLDL (if any), ... etc.

Reply: Thank you for your comments and suggestions. Table 3 (and table 4) present all the values available in the literature for cholesterol, which are total cholesterol, HDL-cholesterol and LDL-cholesterol.

4-Some clarifications of the mode of actions required to highlight the effects (physiological mechanisms) in addition to the impacts (the end result).

Reply: Thank you for your insightful query regarding the need for further clarification on the modes of action of Spirulina, highlighting the physiological mechanisms in addition to the impacts observed in our study. We appreciate the opportunity to elaborate on this aspect in the revised version of the manuscript.

5-Subtitles should be informative (revise the attached file).

Reply: Revised according to the attached file provided.

6-Blood health markers require sub-categories such as Lipid profile, antioxidants, oxidative stress markers, immune indicators, in some cases clarify either normal or stress conditions .... etc.

Reply: Thank you for your comments and suggestions. We sub-categorised the blood health markers in Table 3 in lipid profile, antioxidants, and oxidative stress markers and clarified either normal or stress conditions. The discussion is also separated into paragraphs according to this separation.

7-References require extensive revision according journal format. Try to search more I think there are some beneficial research work could be find.

Reply: Thank you for highlighting the need for an extensive revision of the references according to the journal's format. We appreciate the opportunity to improve this aspect of our manuscript.

Reviewer 2 Report

Comments and Suggestions for Authors

The manuscript is well written. It was well organized and was clear on points. Minor edit suggestions below.

LINE

45 - needs a closed parenthesis

62 - Replace "So," with, "Lastly,". This will help serve as a transition sentence for the next paragraph.

79 - Replace "probably" with "likely"

124 - Should read as "rate of gain"

215 - Remove "as a" Should read, "while Spirulina demonstrates potential beneficial dietary supplements...

TABLES 1, 2 & 3 - Perhaps organizing tables in a specific order to show a trend of different effects. This could better tell a story.

Author Response

Reviewer 2

The manuscript is well written. It was well organized and was clear on points. Minor edit suggestions below.

Reply: Thank you for your comments and suggestions. We appreciate it and try to address all of them.

LINE

45 - needs a closed parenthesis

Reply: Thank you for your comments. Closed.

62 - Replace "So," with, "Lastly,". This will help serve as a transition sentence for the next paragraph.

Reply: Thank you for your suggestion. Replaced.

79 - Replace "probably" with "likely"

Reply: Thank you for your suggestion. Replaced.

124 - Should read as "rate of gain"

Reply: Thank you for your suggestion. Replaced for “body weight gain”.

215 - Remove "as a" Should read, "while Spirulina demonstrates potential beneficial dietary supplements...

Reply: Thank you for your suggestion. Replaced.

TABLES 1, 2 & 3 - Perhaps organizing tables in a specific order to show a trend of different effects. This could better tell a story.

Reply: Thank you for your suggestion. We understand the importance of organizing tables to better illustrate the trends and impacts of Spirulina supplementation. Currently, Tables 1, 3 and 5 detail the raw data used to analyse the correlations between cumulative Spirulina intake and various parameters, while Tables 2, 4 and 6 present the results of these correlation analyses. In each table, the variables are organized to follow a conventional narrative flow that reflects the progression of effects and in crescent order for cumulative algae intake. For example, in the performance table, data is presented in the following order: final body weight, body weight gain and feed conversion ratio. This structure is intended to provide a clear and logical progression from raw data to analysed results, thereby enhancing the reader's understanding of the study's findings. We will review the organization once more to ensure it effectively tells the story of Spirulina's impacts.